# GFI-Net: Global Feature Interaction Network for Monocular Depth Estimation

**DOI:** 10.3390/e25030421

**Published:** 2023-02-26

**Authors:** Cong Zhang, Ke Xu, Yanxin Ma, Jianwei Wan

**Affiliations:** College of Electronic Science and Technology, National University of Defense Technology, Changsha 410073, China

**Keywords:** monocular depth estimation, global attention mechanism, Transformer block, multi-scale feature extraction

## Abstract

Monocular depth estimation techniques are used to recover the distance from the target to the camera plane in an image scene. However, there are still several problems, such as insufficient estimation accuracy, the inaccurate localization of details, and depth discontinuity in planes parallel to the camera plane. To solve these problems, we propose the Global Feature Interaction Network (GFI-Net), which aims to utilize geometric features, such as object locations and vanishing points, on a global scale. In order to capture the interactive information of the width, height, and channel of the feature graph and expand the global information in the network, we designed a global interactive attention mechanism. The global interactive attention mechanism reduces the loss of pixel information and improves the performance of depth estimation. Furthermore, the encoder uses the Transformer to reduce coding losses and improve the accuracy of depth estimation. Finally, a local–global feature fusion module is designed to improve the depth map’s representation of detailed areas. The experimental results on the NYU-Depth-v2 dataset and the KITTI dataset showed that our model achieved state-of-the-art performance with full detail recovery and depth continuation on the same plane.

## 1. Introduction

Monocular depth estimation techniques are inspired by the fact that the human eye can easily deduce the approximate size of objects, their relative position, and even the relative distance from the eye to the target. Monocular depth estimation based on deep learning is computationally simple with a low hardware cost, unlimited scenarios, and no need for stereo matching. It can be widely used in areas such as intelligent driving, heritage reconstruction, and intelligent unmanned combat. However, there are still several problems, such as insufficient estimation accuracy, inaccurate localization of details, and depth discontinuity in planes parallel to the camera plane. For example, the depth of the wall parallel to the camera plane in the depth map is discontinuous, and the edges of the target are blurred. To solve these problems, we propose the Global Feature Interaction Network (GFI-Net), which aims to utilize geometric features, such as object locations and vanishing points, on a global scale.

In monocular depth estimation networks, the full use of global and local information allows for the accurate recovery of detailed areas of the depth map, improved accuracy of the depth map, and accurate recovery of the same depth plane [1,2,3]. Many geometric features in the scene, such as the location of targets and vanishing points, are necessary for the network’s global understanding of the scene [3]. Therefore, the global information of the image needs to be preserved in the monocular depth estimation network to obtain a highly accurate depth map. However, previous approaches neglected the use of the interaction information of the three dimensions of the feature map height, width, and channel to improve the global information of the depth estimation network [3,4,5]. Therefore, we developed a new Global Interaction Attention Mechanism (GIAM) module to boost the performance of deep estimation networks by mitigating information loss during encoding and amplifying global interactions.

The main architecture of a monocular depth estimation network based on deep learning is an encoder–decoder network. The encoder usually migrates the backbone network for image classification, and the decoder aggregates the features extracted by the encoder to recover a high-precision depth map. The convolution-based backbone network continuously down-samples the input scene image to extract features at different scales of the image. Continuous down-sampling reduces the computational effort of the network and increases the perceptual field of the network. However, this operation greatly compresses the resolution of the image, causing the encoder to inaccurately locate the target edges when recovering the depth map. The Transformer block is used as the basis for the encoder in order to avoid compressing the image resolution and reducing computational losses while preserving the network perception field. The experimental results showed that the use of Transformer blocks instead of traditional convolution as encoders is competent for dense prediction tasks. Finally, the paper used a highly utilized decoder to reduce the amount of computation and to recover the depth map accurately [3].

Our main contributions in this paper are as follows:(i)The Global Feature Interaction Network (GFI-Net) was designed in which the geometric information of the scene is fully utilized. The use of geometric information helps to construct fine-grained depth maps. In addition, the depth of the plane parallel to the camera is continuous.(ii)The Global Interactive Attention Mechanism (GIAM) was designed to improve the accuracy of the depth map, guaranteeing the same depth at the same distance from the camera plane. It fully preserves the network’s channel and spatial information, enhances the interaction between the three dimensions of the feature map width, height, and channel, and mitigates information loss.(iii)Using the Transformer block as an encoder avoids the loss of information from continuous down-sampling and increases the accuracy of depth estimation while improving the perceptual field of the network. The global–local fusion module was designed to efficiently fuse low-level pixel information with high-level semantic information to build a fine-grained depth map and connect a low-complexity decoder.(iv)The experimental results on the NYU-Depth-v2 and KITTI datasets showed that our network model achieved state-of-the-art performance, achieving depth continuity in planes parallel to the camera and recovering more complete details. Testing in real scenarios demonstrated the network’s good generalization.

## 2. Related Work

Monocular depth estimation is the recovery of the depth information of a scene. It is divided into traditional methods and methods based on deep learning. The most representative of the traditional methods is the Structure From Motion (SFM) [6]. SFM [6] is used to calculate and recover the 3D information of a scene and the corresponding camera parameters by taking multiple images of the same scene. However, SFM [6] restores blur to areas such as repeated textures and material transparency in depth and is very computationally intensive. A deep-learning-based approach can effectively solve these problems. Traditional codec structures [7], depth regression networks [8,9], and depth classification networks [5,10,11] are the three basic deep learning methods for monocular depth estimation. Eigen [7] designed a depth estimation network model with a coarse-to-fine optimization. The model consists of two convolutional neural networks connected in series, where the coarse estimation network aims to extract the global information about the image. The coding characteristics of the convolutional neural network greatly improve the performance of the depth estimation network [7,10,12] and can be better extended to other tasks. In addition, BTS [1] has suggested a local planar guidance layer that outputs plane coefficients and then uses them in the full-resolution depth estimation. The post-optimization network uses the local information to refine the coarse depth prediction results. Ning [9] proposed a unified autoregressive coder–decoder model that can handle multiple visual tasks at the same time. AdaBins [5] is a typical depth class network that defines the monocular depth estimation problem as a classification task, dividing the depth values adaptively into a number of units and obtaining high-performance results. Agarwal [11] proposed a new strategy to predict the depth as a problem of pixel query refinements.

The Transformer network was first proposed for Natural Language Processing (NLP) tasks, which is also utilized in the field of computer vision currently. To overcome the limitations of Recurrent Neural Networks (RNNs) in natural language processing, Vaswani et al. [13] proposed a self-attentive mechanism with a Multi-Layer Perceptron (MLP). A Vision Transformer (ViT) [14] was initially developed in the realm of computer vision to handle the difficulties of image classification. However, few researchers have attempted to solve the depth estimation task using the Transformer block as a backbone. AdaBins [5] used a minimized version of the vision Transformer to estimate depth intervals adaptively. Vision Transformers for Dense Prediction (DPT) [4] employ a ViT to enlarge the perceptual area and then couple a convolutional-neural-network-based decoder for dense prediction. However, their network parameters are too large. In addition, the DPT’s training requires an additional dataset. Agarwal [11] used the encoder backbone based on the Swin Transformer to improve efficiency. In contrast, our encoder reduces complexity while retaining the receptive field and does not require additional datasets.

Attention mechanisms are derived from the study of human vision. Humans selectively focus on a part of all information and overlook other apparent information due to bottlenecks in information processing. Several researchers have examined how to increase the performance of attention mechanisms in computer vision tasks. The most representative of this work is Squeeze-and-Excitation Networks (SENets). The SENet [15] was the first network to select the important channels using the channel attention mechanism and channelwise feature fusion. However, it is less effective at pixel selection. FcaNet [16] popularized the compression of the channel attention mechanism in the frequency domain and proposed a method of multi-spectral channel attention. Both the spatial and channel aspects are taken into account by a later attention mechanisms. Woo et al. [17] presented the Convolutional Block Attention Module (CBAM), which sets up channel and spatial attention operations sequentially, whereas Park et al. [18] proposed the Bottleneck Attention Module (BAM), which accomplishes this in parallel. Both, however, overlook the channel–spatial interactions, resulting in the loss of cross-dimension information. Therefore, for the monocular depth estimation network, we developed an attention mechanism that can capture the interactions of the three parts: the channel, spatial breadth, and height. This operation allows for the introduction of geometric features, which increase the depth estimate accuracy and depth map detail recovery.

Monocular depth estimation networks based on deep learning also suffer from the following problems:(i)The encoder in the network greatly compresses the resolution of the image when extracting features from the network input, resulting in inaccurate edge localization by the decoder when recovering the depth map, so many methods are investigating the precise location of depth edges with large variations in the depth map. As in Figure 1 [10], from left to right, the scene images, the true values of the depth values, and the estimation results of the existing network model are shown. In Figure 1 [10], the edges of the sofa in the first row, the edges of the bedside table in the second row, and the items on the dining table in the third row appear to have blurred depth edges, as shown in the white boxes. Therefore, the Transformer block is used as an encoder to reduce information loss and improve the accuracy of the depth map.(ii)The existing monocular depth estimation algorithm based on depth learning rarely considers the relative change of pixel information, which will lead to the phenomenon of depth discontinuity in the plane parallel to the camera. As in Figure 2 [10], from left to right, the real picture of the scene, the real value of the depth value, and the depth estimation results of existing methods are shown. In the existing estimation results, the same plane parallel to the camera will have depth discontinuity. As shown in the white box in the Figure 2 [10], the wall in the first row, the blackboard on the back wall in the second row, and the wall in the third row all have depth discontinuities. Therefore, the global attention interaction mechanism was designed in the network to capture the interactive information of the width, height, and channel of the feature map. Interactive information can effectively capture the relative position relationship between pixels and solve the problem of discontinuity at the same depth.

## 3. Methods

### 3.1. Global Feature Interaction Network

The global feature interaction network obtains the depth map $\hat{Y}\in R^{H\times W\times 1}$ from one image $I\in R^{H\times W\times 3}$, which consists of a Transformer encoder, a global interaction attention mechanism, a low-complexity decoder, and a local–global feature fusion module. ($\hat{Y}$ represents the depth map. *I* stands for the image. *R* represents the matrix space. *H* represents the height of the feature map. *W* represents the width of the feature map. *C* represents the number of channels in the feature map.) Figure 3 shows the overall architecture of the network. The Transformer encoder is more suitable for dense prediction. While preserving the global receptive field of the network, the Transformer encoder avoids the problem of inaccurate depth map detail recovery caused by the huge compressed image’s resolution [4]. The global attention interaction mechanism module enables the network to capture the interactive information between the three dimensions of the channel, feature map width, and height and improves the encoder’s ability to extract global features from RGB images. The low-complexity decoder can obtain a high-precision depth map while retaining the low complexity of the network [3]. The local–global feature fusion module fully fuses the low-scale pixel position information and the high-scale semantic information to improve the detail recovery capability of the network. In the next subsections, we go over each of the four proposed modules in depth.

### 3.2. Transformer Encoder

The network uses the Transformer encoder to be able to solve the dense prediction problem. In order to extract the global information of RGB images in the network, the encoder used in this section uses layered Transformer-based blocks to expand the receptive field. It avoids the problem of continuous down-sampling greatly compressing the resolution and leading to inaccurate detail edge recovery when the decoder recovers the depth map. The network input image pixel size is 680 × 480. Then, it is embedded into the neural network in block mode through a 3×3 convolution operation. The patches are utilized as the Transformer block’s inputs. Each Transformer block consists of three modules: self-attention operation, MLP-Conv-MLP, and patch merging. We used four Transformer blocks, and each block generates $\frac{1}{4},\frac{1}{8},\frac{1}{16}$, and $\frac{1}{32}$ scale features with $\left[C_{1},C_{2},C_{3},C_{4}\right]$ dimensions. Following successive Transformer encoding, multi-scale features are obtained, with the lower-level features rich in pixel location information and the higher-level features rich in semantic information of the image. Fusing pixel location information and semantic information of the scene is important for the depth estimation task detail recovery problem.

### 3.3. Global Interaction Attention Mechanism

The depth estimation task suffers from the problem of depth discontinuities occurring in the same plane at the same distance. To solve this problem, a global attention mechanism is designed to reduce the information leakage during encoding and to expand the global information interaction. The proposed global attention mechanism captures important features of three parts, namely channel, spatial width, and height, and enhances the features of the interaction across the parts. We redesigned the submodule using the CBAM’s sequential channel and spatial attention mechanism [17]. The entire procedure is seen in Figure 4 and is detailed in Equations (Equation 1) and (Equation 2) [17]. The features from the encoder are defined as $F_{1}\in R^{C\times H\times W}$, and the intermediate state outputs from the GAM are $F_{2}$ and $F_{3}$, respectively.
(1)$$F_{2}=M_{c}\left(F_{1}\right)⊗F_{1}$$
(2)$$F_{3}=M_{s}\left(F_{2}\right)⊗F_{2}$$
where $M_{c}$ represents the operation of channel attention on the feature map and $M_{s}$ represents the operation of spatial attention on the feature map. ⊗ stands for multiplication between elements.

The **channel attention module** in the depth estimation network uses 3D permutation to retain three dimensions of information. The spatial dependencies across dimensions are then amplified using a two-layer MLP. MLP is an encoder–decoder architecture that employs the *r* reduction ratio of the BAM [18]. Attention operations in the channel dimension help the encoder capture high-level semantic information in the scene so that the depth estimated by the depth estimation network is continuous in a plane parallel to the camera.

The **spatial attention module** in the depth estimation network use two convolutions, which fuse the features in the spatial dimension. In this part, we set the reduction ratio to equal *r*, which is the same as the BAM. The max-pooling operation was removed in this part in order to keep the feature map and reduce the information loss caused by max-pooling during encoding. However, the spatial attention module can occasionally increase the number of parameters in the model considerably. Therefore, we employed group convolution with channel shuffle [19] in our depth estimation network to prevent a major rise in parameters.

The illustrations the two modules are shown in Figure 5.

### 3.4. Lightweight Decoder

Image *I* is passed through the encoder to obtain a feature map of size $\frac{1}{32}H\times \frac{1}{32}\;W\times C_{4}$. To reconstruct the depth map, our network utilizes a decoder [3], which is lightweight and efficient. It is used in order to make the feature map bm $F_{3}$’s size become H×W×1. To recover the original size, most earlier work stacked numerous bilinear up-sampling layers with convolution or deconvolution layers. However, this model performs better with decoders that have fewer convolution and bilinear up-sampling layers. The 1 × 1 convolution reduces the channel dimension to $N_{c}$ and decreases the computational complexity. The features are then scaled up to $H\times W\times N_{c}$, utilizing continuous bilinear up-sampling. Finally, to obtain the expected depth map, we used two convolutional layers and a softmax layer.

### 3.5. Local–Global Feature Fusion Module

The integration of local and global information is essential to the task of depth estimation. By obtaining an attention map for each feature, we used a fusion module to self-adaptably select and combine local and global information [3]. Figure 6 depicts the detailed structure of the feature fusion module. Firstly, in order to fuse the local and global information, both $F_{D}$ and $F_{E}$ reduce the channel dimensions to 64 by the channel reduction module. Secondly, the global and local features are concatenated in the channel dimension. Thirdly, the spliced features are subjected to two concatenated Conv-BatchNorm-ReLU operations, followed by a Conv-Sigmoid operation to obtain the channel attention map. Fourthly, the obtained channel attention map is multiplied with the original global information and local information to obtain $H_{D}^{i}$. $H_{D}^{i}$ represents the feature after local feature and global feature fusion. To reinforce local continuity, we avoided reducing the scale feature’s size to 1/4. Fusing pixel location information and semantic information of the scene is important for the depth estimation task detail recovery problem.

### 3.6. Loss Function

In order to calculate the loss between the predicted depth map $\hat{Y}$ and the ground truth depth map Y, a scale-invariant log-scale loss was used to train the model. The specific formula is as follows:(3)$$L=\frac{1}{n}\sum_{i}d_{i}^{2}-\frac{1}{2n^{2}}\left(\sum_{i}d_{i}^{2}\right)$$
where $d_{i}=\log y_{i}-\log y_{i}^{*}$. $y_{i}^{*}$ and $y_{i}$ represent the i-th pixel in $\hat{Y}$ and Y. n is the number of pixels.

## 4. Experiments

The experiments in this section were performed by training and testing on the NYU Depth V2 and KITTI datasets, the most widely cited monocular depth estimation datasets at present. The depth estimation accuracy results obtained in the experiment were compared and analyzed. The results of depth estimation were also visualized and analyzed. Ablation experiments were conducted on the global attention mechanism module of the network to verify the effectiveness of each component. Finally, it was tested on a real scene to verify the generalization of the network model.

### 4.1. Dataset

The **NYU Depth V2** [20] dataset is an indoor scene dataset with depth ground truth values for 3D scene understanding, as shown in Figure 7a. The dataset comprises 464 distinct scenes taken by a Microsoft Kinect in various buildings, of which 249 were utilized for training and 215 for testing. The dataset, which is a subset of the NYU dataset with segmentation labels, consists of 1449 picture pairings, of which 795 were utilized for training and the remaining 654 for depth estimation testing. The photos have a resolution of 640 × 480.

The **KITTI** [21] dataset was gathered outdoors utilizing a transportable in-vehicle platform, as illustrated in Figure 7b. The RGB images were captured by stereo-aligned and corrected cameras. A revolving Velodyne laser scanner placed on the driving vehicle captured a depth map. The KITTI dataset contains data collected from genuine photographs in the “city”, “residential”, and “road” categories. The KITTI dataset’s 56 scenes were separated into two parts: 28 for training and 28 for testing. The depth maps’ ground truth was created by projecting the 3D points from the LiDAR onto the left RGB camera using the supplied intrinsic and extrinsic parameters. Because the 3D points are not dense enough, the resulting pictures are quite sparse. The photos have a resolution of 1224 × 368.

### 4.2. Implementation Detail

The network model we built was implemented on the pytorch framework. In the training phase, the model used the one-cycle learning rate strategy, and the optimizer was Adam. The number of epochs was set to 25. Batch_size was set to 4. In the encoder stage, the pre-training parameters of MiT-b4 were migrated to reduce the training time. Initialize the encoder using the parameters of MiT-b4 [22]. Set the values of $F_{E}^{1}$ to [64×112×114]. Set the values of $N_{C}$ to 64. The size of the convolution kernel was [3×3]. The hardware used an NVIDIA 2080ti GPU. The Ubuntu system version was 18.04. In order to facilitate fair comparison, use Eigen’s predefined center clipping on the NYU Depth V2 dataset for evaluation. The maximum range of set depth was 10 m. Use the clipping defined by Garg on the KITTI dataset, and set the maximum range of depth estimation to 80 m.

A widely recognized assessment approach with five evaluation criteria was presented in [23] for evaluating and comparing the performance of various depth estimation networks. The formula for the assessment model is specified as follows:(4)$$\mathrm{RMSE}=\sqrt{\frac{1}{\left|N\right|}\sum_{i=N}{∥d_{i}-d_{i}^{*}∥}^{2}}$$
(5)$$\mathrm{RMSE}\log=\sqrt{\frac{1}{\left|N\right|}\sum_{i=N}{∥\log\left(d_{i}\right)-\log\left(d_{i}^{*}\right)∥}^{2}}$$
(6)AbsRel=1|N|∑i∈Ndi−di∗di∗
(7)$$\mathrm{Accuracies}:\max\left(\frac{d_{i}}{d_{i}^{*}},\frac{d_{i}^{*}}{d_{i}}\right)=E<\mathrm{thr}$$
where $d_{i}$ is the predicted depth value of the pixel i and $d_{i}^{*}$ denotes the true value of the depth. N is the total number of pixels with a real depth value and thr is the threshold value.

### 4.3. Comparison with the Latest Methods

First, the depth estimation effects of different algorithms were compared on the NYU Depth v2 dataset. The quantitative experimental results are shown in Table 1. DPT requires additional training datasets. Some examples of the estimation results are shown in Figure 8. It can be seen from Table 1 that the algorithm in this section achieved the optimal results on three indicators, $\delta_{i}$, RMSE, and $\log_{10}$. Specifically, the optimal results of 0.908, 0.347, and 0.042 were obtained on the indicators $\delta_{i}$, RMSE, and $\log_{10}$, which proved the effectiveness of the algorithm in this section. Compared with the algorithm in [3], the algorithm in this paper improved 0.1% on index $\delta_{i}$. The error of the RMSE indexes reduced by 0.8%. In terms of network structure, compared with [3], the algorithm in this section added a global attention interaction mechanism in the network to reduce information leakage in the coding process and expand the interaction of the channel, space width, and height. Compared with [4], the parameter decreased by 83.5%, increased by 0.4% on index $\delta_{i}$, and decreased by 2.8% and 4.5% on indices RMSE and $\log_{10}$, respectively. Compared with GLPD [3], we added a global interaction mechanism that can capture the interaction information of the width, height, and channel of the feature graph in the network. At the same time, a lightweight encoder was migrated, which integrates the location information of the bottom pixel and the semantic information of the top layer, reducing the loss of the network and ensuring the effectiveness of the algorithm.

Some estimation results are shown in Figure 8. The first column in the figure is the RGB image of the experimental scene, and the following four columns from left to right are from [3,4], the depth estimation results of the algorithm in this paper, and the ground truth. As indicated by the box in the figure, this area is more complex in the scene. For the blackboard marked with the first line of boxes, only the visualization results of this algorithm clearly show the triangles on the blackboard. For the position of the door frame marked by the box in the second line, only the algorithm in this paper explicitly learned the edge of the door frame and the depth of the transparent area in the scene. For the ladder area marked by the box in the third line, only the algorithm in this paper can learn the specific contour of the lifebuoy on the ladder. For the bookcase in the living room marked by the fourth line of boxes, only the algorithm in this paper can learn the edge contour of the bookcase, and the information in the scene was most accurate. In the rear area marked by the fifth line box, only the algorithm in this paper explicitly learned the distance of the object’s relative position. In the scene marked with the sixth line of boxes, only the algorithm in this paper clearly learned the depth of the wall; there was no fault, and the edge of the box under the bed in the scene was clear. The global feature interaction network model algorithm enhanced the interaction between local information and global information and enhanced the interaction between the width, height, and channel of the feature map, which made the network learn more fully about the target location of the scene and restore the details accurately.

Secondly, the depth estimation effects of different algorithms were compared on the KITTI dataset. The quantitative experimental results are shown in Table 2. The DPT requires additional training datasets. Some examples of the estimation results are shown in Figure 9. It can be seen from Table 2 that the algorithm in this paper achieved the best results on the indicators $\delta_{i}$, RMSE, and $\log_{10}$ for the KITTI dataset. Specifically, the optimal result of 0.996 was obtained on index $\delta_{i}$; 2.303 was obtained on index RMSE; 0.087 was obtained on index $\log_{10}$; these prove the effectiveness of the algorithm. Compared with the algorithm in [3], the algorithm in this paper improved the $\delta_{i}$ index by 3.4%. The error of RMSE index was reduced by 0.8%. In terms of network structure, compared with [3], this algorithm added a global attention interaction mechanism in the network to reduce information leakage in the coding process and expand the interaction of the channel, space width, and height. Compared with [4], the parameter decreased by 83.5%, increased by 3.8% on index $\delta_{i}$, and decreased by 10.4% and 5.4% on indices RMSE and $\log_{10}$, respectively. On the KITTI dataset, adding the global interaction mechanism also had a good effect.

Some estimation results are shown in Figure 9. The first column in the figure is the RGB image of the experimental scene, and the following four columns from left to right are [3,4], the depth estimation results of the algorithm in this chapter, and the ground truth. It is not difficult to see that the details of the estimation results of this algorithm were restored accurately. For example, objects such as cars, branches, and window frames were clearly delineated in the scene. Therefore, it can be concluded that the addition of the global attention mechanism was conducive to the restoration of the depth map details.

### 4.4. Real Scene Test Results and Analysis

The trained network model was tested on real scenes with good results. The training model was tested on the home environment, school environment, office environment, and road environment, respectively. The home environment took the bedroom and restaurant as the test environment.

The bedroom environment is shown in Figure 10. The training results obtained on the NYU Depth dataset can be well extended to the home environment. The position of the target in the scene can be clearly distinguished in the visualization results. As shown in Figure 10, you can clearly learn the outlines of the TV cabinets, chairs, tea tables, beds, potted plants. and bedding. It has very rich details. In the dining room environment, as shown in Figure 11, you can clearly learn the outline of the people and the table in the scene, and the depth is continuous in the plane parallel to the camera.

The school environment takes the conference room, classroom, and library environment as the test environment. Visualize the result of the depth estimation, which can clearly restore the contour of the target in the scene. It is helpful to understand the scene. Specifically, the conference room results are shown in Figure 12, and the classroom environment is shown in Figure 13. It can clearly restore the outline of the chairs and tables in the scene. The library environment is shown in Figure 14. Compared with the former two, the library environment is more complex. The books in the bookcase have more details. The test results in this paper obtained clear outlines.

Take the school gate, the road in front of the library, and the dormitory building as examples for the outdoor environment. The visualization results are shown in Figure 15. The depth map results obtained have clear object boundaries. Specifically, the school gate, vehicles running on the road, and students walking on the road have clear edge contours.

## 5. Ablation Study

In this section, we performed ablation experiments on the publicly available datasets, NYU-Depth V2 and KITTI, to validate the effectiveness of our method. The roles of the spatial and channel attention mechanisms in the network model were first assessed separately. To better understand the contribution of spatial and channel attention, separate ablation experiments were conducted, where only the spatial attention module or the channel attention module was used in the network. ch indicates that only the channel attention module was added. sp indicates that only the spatial attention module was added. Table 3 shows the results of the ablation experiments on the NYU Depth V2 dataset, and Table 4 shows the results of the ablation experiments on the KITTI dataset. Experiments on the NYU Depth V2 and KITTI datasets showed that the best results of GAM(sp+ch) were 0.347 and 2.303 on index RMSE. In addition, by observing the important indicator of the RMSE, it was found that ch and sp alone had little effect on system performance improvement. Only by capturing the attention of the channel and space at the same time can the performance of the depth estimation be improved. Compared with the original algorithm, our proposed algorithm improved the RMSE index by 0.8%. The network can capture the interaction information in three dimensions: channel information, feature map width, and height information by adding the global attention mechanism module. The results showed that this operation obtained the highest accuracy of depth map estimation.

## 6. Conclusions

In this paper, the global feature interaction network was proposed to enhance the recovery of depth information for detailed regions and to obtain highly accurate depth maps. The interaction information between the three dimensions, the channel information, height, and width of the feature map, allowed for better recovery of detailed areas and a continuity of depth over large planes at long distances. More geometric features, such as object locations and vanishing points, were utilized on a global scale. The model achieved state-of-the-art accuracy on both the NYU Depth v2 dataset and the KITTI dataset. It also outperformed existing methods in terms of detail recovery with the low complexity of the model in terms of the number of parameters. At the same time, it was tested on real scenes, which proved that the network has good generalization.

## Figures and Tables

**Figure 1 entropy-25-00421-f001:**
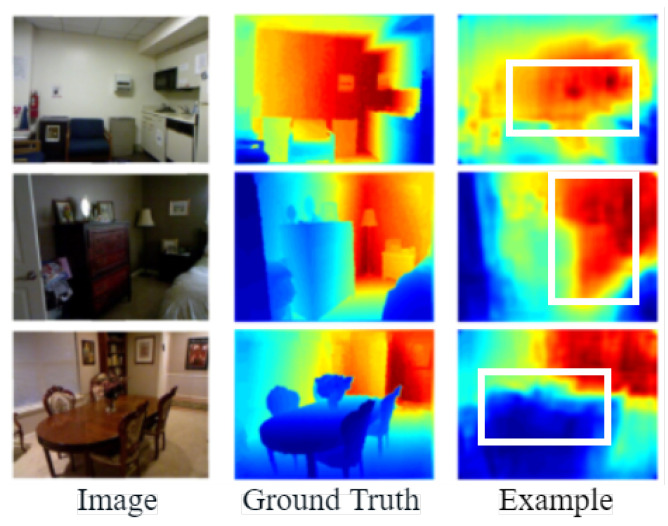
Example of inaccurate edge positioning [10].

**Figure 2 entropy-25-00421-f002:**
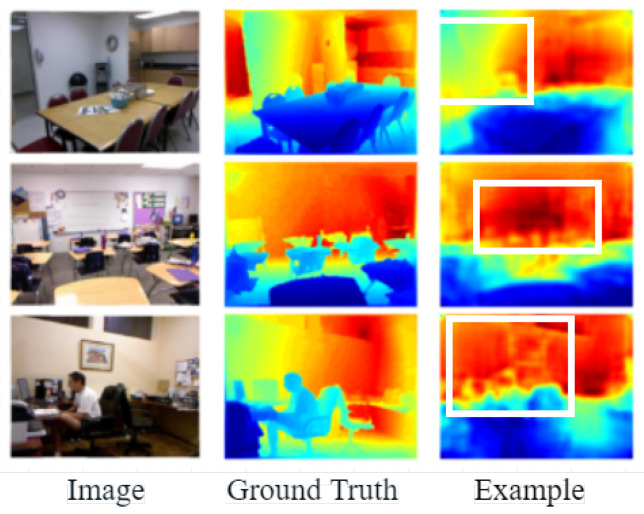
Examples of discontinuities in the same plane depth [10].

**Figure 3 entropy-25-00421-f003:**
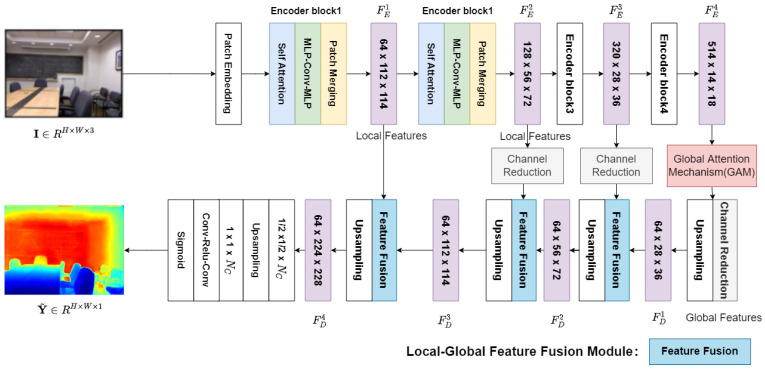
The architecture of the proposed global feature interaction network consists of four components: an Transformer encoder, a global interaction attention mechanism, a decoder, and a feature fusion module.

**Figure 4 entropy-25-00421-f004:**
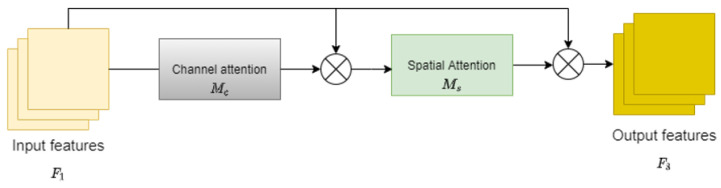
The illustration of GAM module.

**Figure 5 entropy-25-00421-f005:**
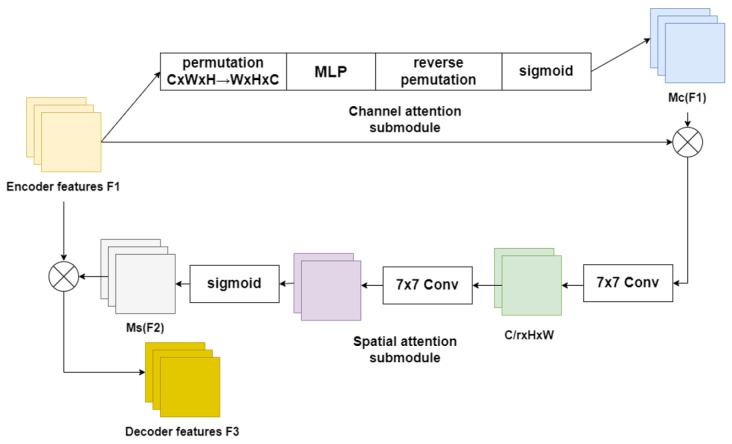
The illustration of the channel and spatial attention modules.

**Figure 6 entropy-25-00421-f006:**
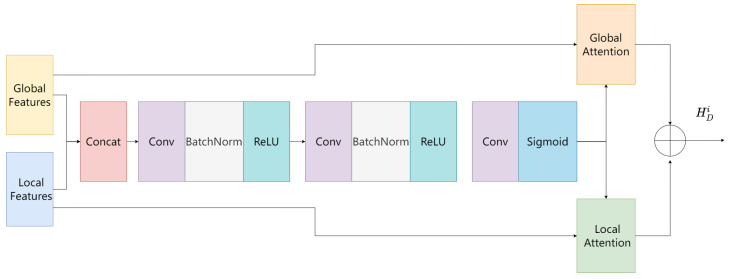
Detailed description of the local–global feature fusion module.

**Figure 7 entropy-25-00421-f007:**
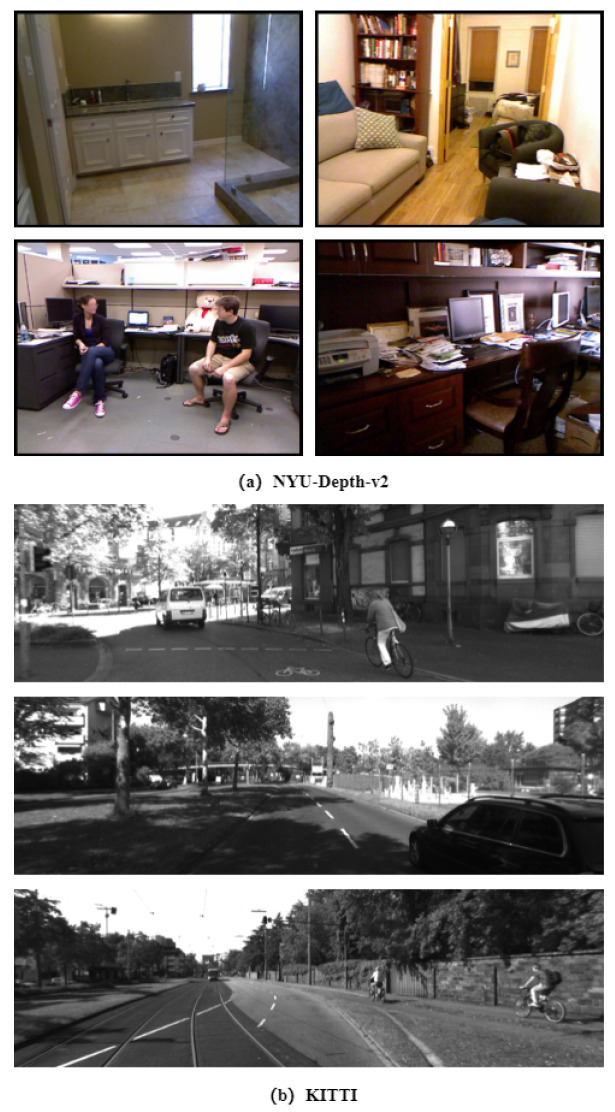
Sample images of (**a**) NYU-Depth-v2 and (**b**) KITTI.

**Figure 8 entropy-25-00421-f008:**
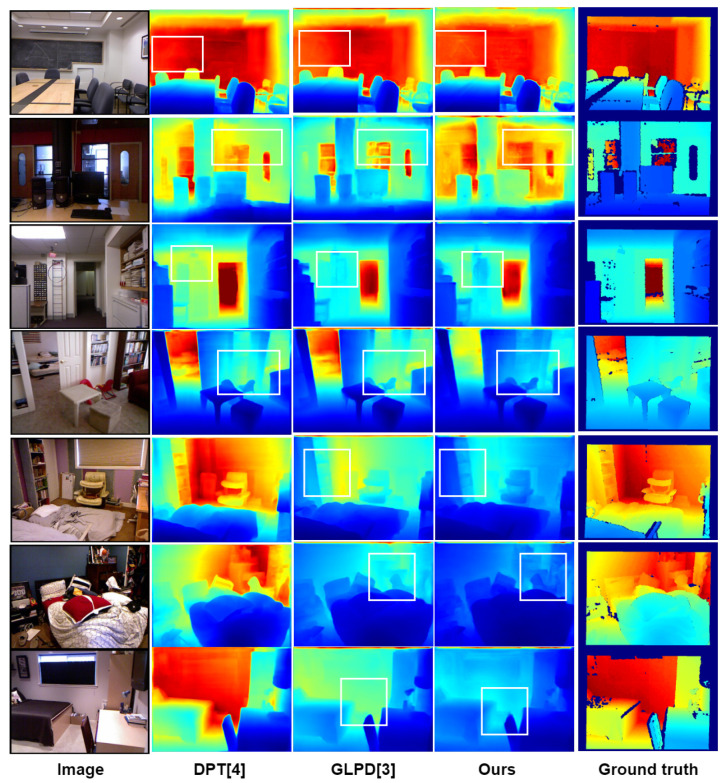
Visualization of the effect on the NYU-Depth v2 dataset. The white boxes in the diagram refer to areas where detail recovery is better than other works.

**Figure 9 entropy-25-00421-f009:**
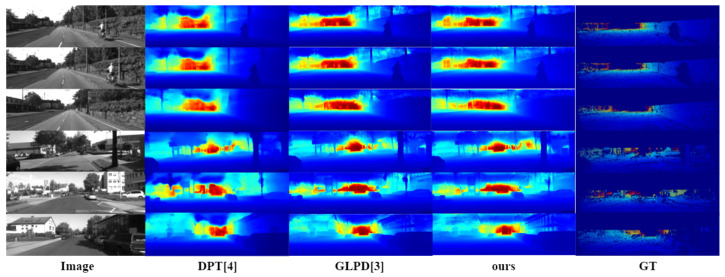
Visualization of the effect on the KITTI dataset.

**Figure 10 entropy-25-00421-f010:**
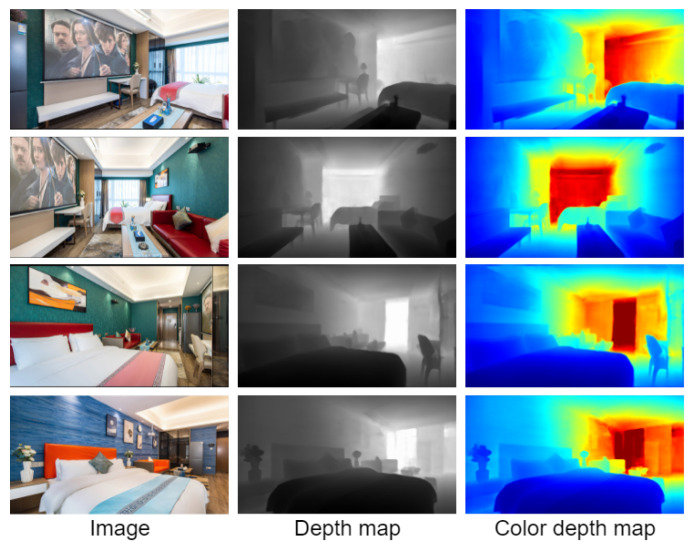
Visualization of the effect on the bedroom environment.

**Figure 11 entropy-25-00421-f011:**
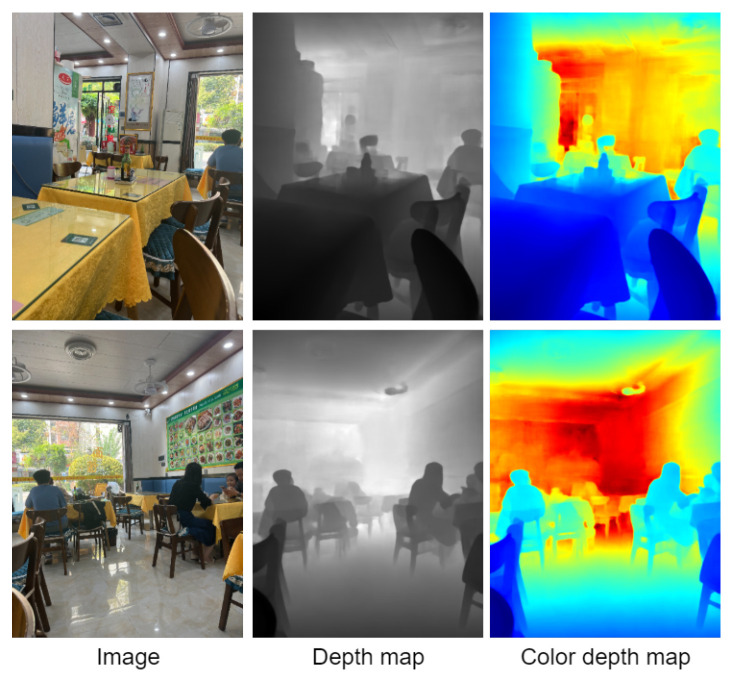
Visualization of the effect on the dining room environment.

**Figure 12 entropy-25-00421-f012:**
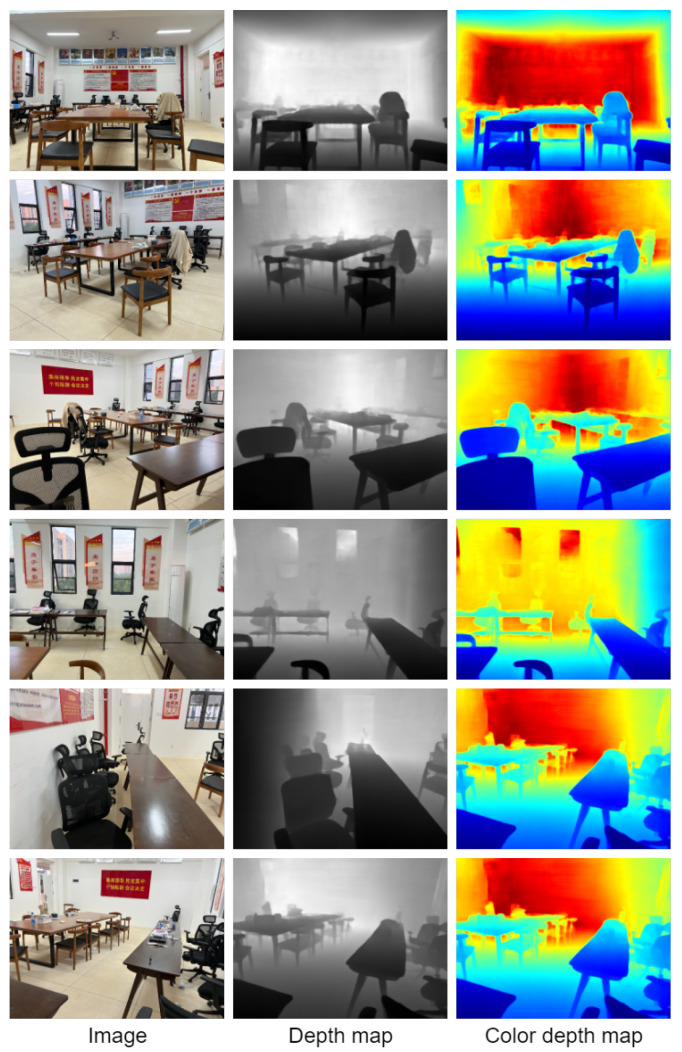
Visualization of the effect on the conference room environment.

**Figure 13 entropy-25-00421-f013:**
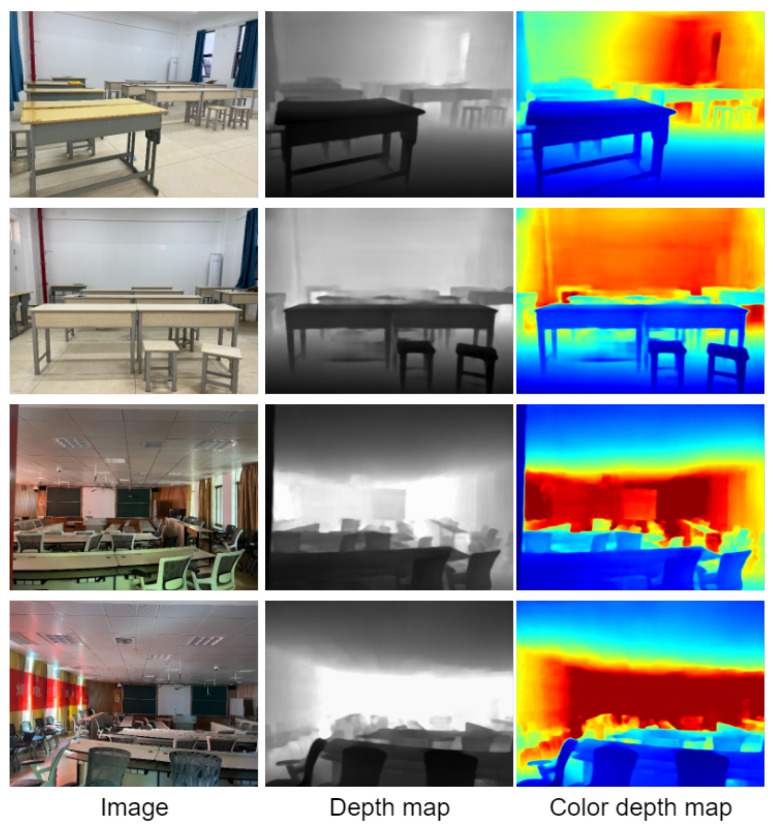
Visualization of the effect on the classroom environment.

**Figure 14 entropy-25-00421-f014:**
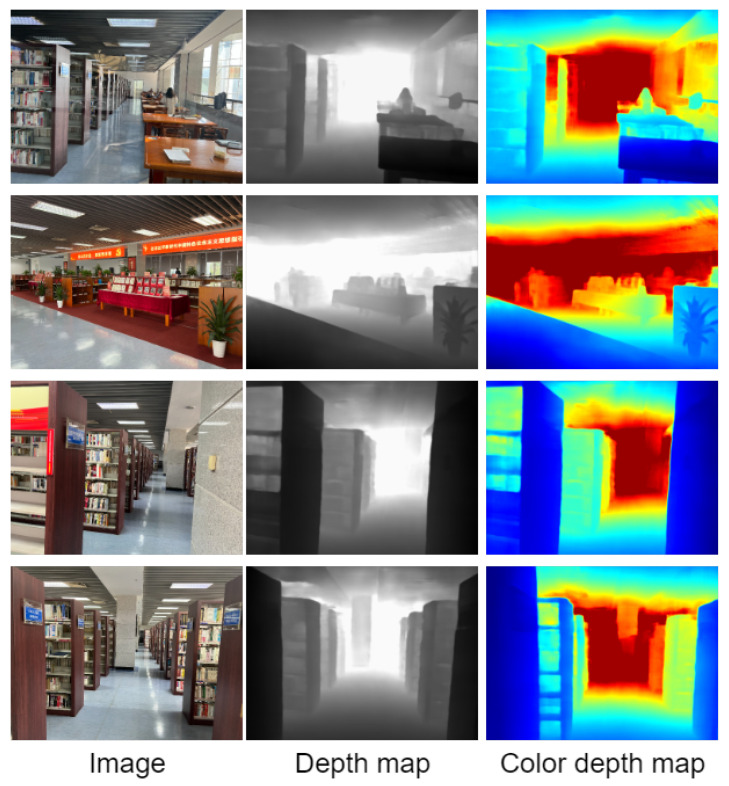
Visualization of the effect on the library environment.

**Figure 15 entropy-25-00421-f015:**
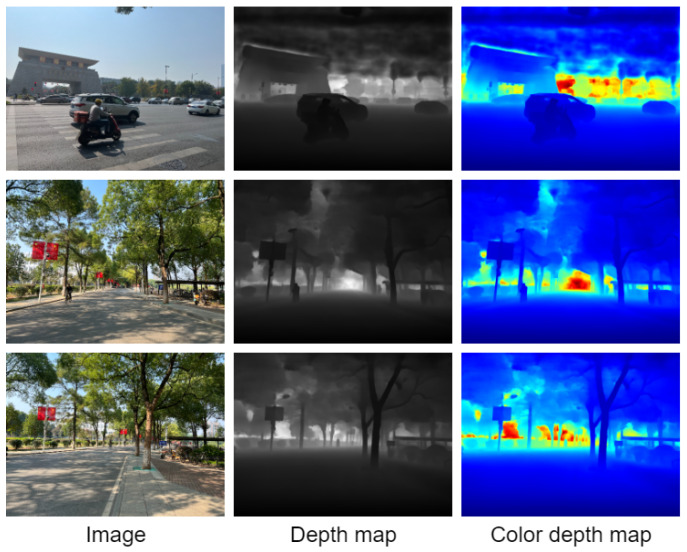
Visualization of the effect on the outdoor environment.

**Table 1 entropy-25-00421-t001:** The accuracy of the proposed model on the NYU-Depth v2 dataset. Additional datasets are required to train the DPT model.

Method	Params (M)	$\delta_{1}↑$	$\delta_{2}↑$	$\delta_{3}↑$	AbsRel↓	RMSE↓	$\log_{10}↓$
Eigen [7]	141	0.769	0.950	0.988	0.158	0.641	-
Fu [10]	110	0.828	0.965	0.992	0.115	0.509	0.051
Yin [12]	114	0.875	0.976	0.994	0.108	0.416	0.048
DAV [24]	25	0.882	0.980	0.996	0.108	0.412	-
BTS [1]	113	0.885	0.978	0.994	0.110	0.392	0.047
AdaBins [5]	78	0.903	0.984	0.997	0.103	0.364	0.044
DPT [4]	123	0.904	**0.988**	**0.998**	0.110	0.357	0.045
GLPD [3]	62	0.907	0.986	0.997	**0.098**	0.350	**0.042**
Ours	67	**0.908**	0.986	0.997	0.100	**0.347**	**0.042**

**Table 2 entropy-25-00421-t002:** The accuracy of the proposed model on the KITTI dataset. Additional datasets are required to train the DPT model.

Method	Params (M)	$\delta_{1}↑$	$\delta_{2}↑$	$\delta_{3}↑$	AbsRel↓	RMSE↓	$\log_{10}↓$
Eigen [7]	141	0.702	0.898	0.967	0.203	0.6.307	0.282
Fu [10]	110	0.932	0.984	0.994	0.072	2.727	0.120
Yin [12]	114	0.938	0.984	0.998	0.072	3.258	0.117
BTS [1]	113	0.956	0.993	0.998	0.059	2.756	0.088
DPT [4]	123	0.959	0.995	**0.999**	0.062	2.573	0.092
AdaBins [5]	78	0.964	0.995	**0.999**	0.058	2.360	0.088
GLPD [3]	62	0.963	**0.995**	**0.999**	0.059	2.322	0.089
Ours	67	**0.996**	0.966	**0.999**	**0.057**	**2.303**	**0.087**

**Table 3 entropy-25-00421-t003:** Ablation studies on NYU Depth V2.

Method	$\delta_{1}↑$	$\delta_{2}↑$	$\delta_{3}↑$	AbsRel↓	RMSE↓	$\log_{10}↓$
GAM(ch)	0.904	0.985	0.996	0.103	0.352	0.043
GAM(sp)	0.911	0.986	0.997	0.098	0.351	0.042
GAM	0.907	0.986	0.997	0.098	0.350	0.042
GAM(sp+ch)	0.908	0.986	0.997	0.100	**0.347**	0.042

**Table 4 entropy-25-00421-t004:** Ablation studies on KITTI.

Method	$\delta_{1}↑$	$\delta_{2}↑$	$\delta_{3}↑$	AbsRel↓	RMSE↓	$\log_{10}↓$
GAM(ch)	0.964	0.995	0.998	0.060	2.327	**0.085**
GAM(sp)	0.965	0.995	0.998	0.058	2.344	0.089
GAM	0.964	0.995	**0.999**	0.059	2.322	0.089
GAM(sp+ch)	**0.966**	**0.996**	**0.999**	**0.057**	**2.303**	0.087

## Data Availability

Not applicable.

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
