# Peer review of "GFI-Net: Global Feature Interaction Network for Monocular Depth Estimation"

_entropy, 2023, doi:10.3390/e25030421_

Round 1
Reviewer 1 Report
Authors are recommended modifying their manuscript based on the following minor and major comments:
Minor comment:
1-there are a number of typos needed to be addressed; for example line 54: T in Transformer should not be capitalized. line 187.
2- There are several claims for which authors should bring a reference; for instance, the claim in lines 32 and 33 can be mentioned.
Major comments:
1-There are lots of sentences that have been repeated in different parts of the paper. Authors are suggested removing the repeated sentences or at least paraphrasing them. For example, lines 38 to 42 are a matter of repetition. Repeated parts can be replaced by details.
2- Related works are absolutely incomplete and old—most of your references are related to years before 2020. This time gap should be addressed.
3- Please bring the ablation study before the comparative part. Out of the experiments you have in the ablation, grab the best one and then compare that with the rivals.
4- Based on table 3, your proposed method "sp+ch" doesn't outperform GAM sp. What makes matters worse is that the performance of ch is generally worse than the GAM. This means including ch is worthless. Please remove ch and sp + ch and re-write the ablation studies.
5- The proposed method, especially its figures, are a matter of vagueness. Authors are recommended elaborating on different parts of their proposed method. For this purpose, they can get and inspiration from "Spatial Channel Attention for Deep Convolutional Neural Networks". Furthermore, please bring the proposed method in this paper as one of your competitors.
Author Response
Happy New Year to you! Please see the attachment.

Reviewer 2 Report
Page 1:
line 4: What is the novelty of your method? Add it to the abstract.
line 31: explain vanishing points.
line 34: This is a vague statement, add some citations to support your text.
Page 2:
line 40: Add relevant citations
Line 53: " Experimental results show that the use of 5Transformer blocks instead of traditional convolution..." Is this your work or previous work, quite unclear from your statement?
line 55: You discuss computational complexity without mentioning Big O for previous methods and your proposed methods. Add Big O for your method and prove how it is better than the previous ones.
Line 57: The main contributions of the paper are quite vague, the results do not reflect the contributions, and these are merely theoretical statements without any supporting results.
line 80: Add more references for SFM
line 81: What do you mean by textures, explain something about it
line 82: This is again a vague statement "Very computationally intensive" which is not supported. by numbers
Page 3
line 94-95: Grammar problem
line 95: What are the limitations of RNN? How your method has overcome those limitations?
line 100: What is DPT? Explain it here.
line 102: " The disadvantage is that both use convolutional neural 102 networks and transformer modules, which makes the network more complex" This is not a disadvantage at all.
line 129: Add references
Page 4:
line 148, 149: The variables are not explained at all. The mapping equations and mathematical equations need to be rewritten.
Line 163: "Transformer encoder avoids the problem of inaccurate depth map details recovery caused 153 by the huge compressed image resolution." How?
line 168: mention Big O
Page 5
- Eq, 1 and Eq 2: multiplication is shown by two different symbols in the text, be consistent with the mathematical symbols.
- Mathematical variables are not declared and explained in the text
Page 6:
- Same issues with equations and variables, confusing and not explained
Page 7:
- subsection 3.1 and 3.2 are incorrectly labeled, there is nothing in subsection 3.1
Page 8 and Page 9:
Formulas and Eq. are not properly explained.
Page 10:
- What do mean by "log10" , is it the log10 of some number? or some parameter? Correct it in the tables as well
- The improvement of 0.1% is not significant, hence the use of all methods, results become useless. This much improvement could be achieved by tuning the hyperparameters only, we don't need a whole new system.
- Similary, an improvement of 0.8% is not significant.
Author Response
Happy New Year to you!Please see the attachment

Reviewer 3 Report
This manuscript proposes Global Feature Interaction Network to enhance the recovery of depth information for detailed regions and to obtain highly accurate depth maps. The topic is relevant and interesting for the Entropy community. The sentence writing in this manuscript needs further improvement. Here are some of my concerns:
1. The Local-Global Feature Fusion Module is not shown in the overall architecture of the proposed global feature interaction network as shown in Figure 3. You’d better to show each sub-module in the overall architecture, and then introduce the internal structure in detail when the sub-module is introduced.
2. The article does not introduce information on the number of network layers and the size of the convolutional kernel when introducing the global feature interaction network. It is suggested to improve the relevant information to facilitate the reader to reproduce the experiment.
3. The symbols are not described in detail, such as H_D in Figure 6, N in equation (3) and so on.
4. The experimental parameters are not described, such as the patch size, the learning rate and so on.
5. Some of the comparison algorithms are introduced in the introduction or related work section, but some of them are not described. You’d better to describe your comparison algorithm. Moreover, eight comparison algorithms are used on the NYU-Depth v2 dataset. However, six comparison algorithms are utilized on the KITTI dataset. Can you explain the reasons?
6. In Figures 10-15, please label which sub-column represents the depth map results and which sub-column represents the ground truth.
Author Response

(The authors gave the same response as above.)

Round 2
Reviewer 2 Report
Authors have accommodated many of the feedback provided in first round of review, except for a few below:
1. The improvement of 0.1% on index is not very significant, how these results are better than the literature.
2. Please add some statistical test to prove robustness of your results.
Author Response
Thank the experts for their opinions. Please see the attachment

Reviewer 3 Report
The authors have well answered the comments and the quality of the revised manuscript has been improved. I have no more questions and recommend this paper for publication.
Author Response
Thank the experts for their opinions.